# Effects of a Nutraceutical Treatment on the Intestinal Microbiota of Sled Dogs

**DOI:** 10.3390/ani14152226

**Published:** 2024-07-31

**Authors:** Benedetta Belà, Paolo Emidio Crisi, Giulia Pignataro, Isa Fusaro, Alessandro Gramenzi

**Affiliations:** Department of Veterinary Medicine, University of Teramo, Piano d’Accio, 64100 Teramo, Italy; pecrisi@unite.it (P.E.C.); gpignataro@unite.it (G.P.); ifusaro@unite.it (I.F.); agramenzi@unite.it (A.G.)

**Keywords:** nutraceutical, sporting dogs, stress, dysbiosis, intestinal microbiota

## Abstract

**Simple Summary:**

Physical activity is often associated with stress and production of radical oxygen species able to alter the composition of the intestinal microbiota. Usually, sporting dogs present gastrointestinal disorders with a dysbiotic intestinal ecosystem; this alteration is also found in various pathological states. To date, it is not yet clear whether dysbiosis is a cause or a consequence of disease. Given the importance of reducing the damage of an intestinal imbalance as much as possible, we evaluated the effectiveness of a specific nutraceutical product, Microbiotal cane^®^, in limiting the dysbiotic condition found in sporting dogs. The results obtained from the present study show the ability of Microbiotal cane^®^ to maintain the concentration of beneficial bacterial species like *Faecalibacterium* spp., *Turicibacter* spp., *Fusobacterium* spp., and *Clostridium hiranonis*, reducing the increase in the harmful ones like *Streptococcus* spp. and *E. coli* usually found increased in athlete dogs. Additionally, the dysbiosis index reports a value lower than zero, falling within the normobiosis values, which was not observed in dogs that were not taking the nutraceutical product.

**Abstract:**

Dog sledding is the main discipline of working dogs on snow, consisting of a team of dogs pulling a sled under the guidance of the owner. To carry out this sport, dogs must have adequate nutrition and vitamin and antioxidant supplementation to ensure that the physical effort is optimal. The present study evaluated the effect that sporting activity and stress have on the canine intestinal microbiota by dividing the dogs into two groups: a control group that did not take any nutraceutical products and the treated group to which a nutraceutical product was administered. The nutraceutical administered in this study is used in all cases of canine intestinal dysbiosis in which it is essential to quickly restore a balanced intestinal microbiota. The results obtained show that in dogs not taking the nutraceutical, there is an increase in bacteria, such as *Streptococcus* spp. and *E. coli*, considered enteropathogenic to the detriment of beneficial bacterial species such as *Faecalibacterium* spp., *Turicibacter* spp., *Blautia* spp., *Fusobacterium* spp., and *Clostridium hiranonis*. Instead, the group of dogs treated with nutraceutical displays a lower amount of enteropathogenic bacteria and a great increase in the other bacterial species considered beneficial for the animal’s health. The results obtained in the present study show that Microbiotal cane^®^ can be used in dogs subject to intense sporting activity by preventing severe alterations at intestinal ecosystem levels by maintaining intestinal bacterial composition as balanced as possible.

## 1. Introduction

The gastrointestinal microbiota is defined as the set of all microorganisms (bacteria, archaea, fungi, protozoa, and viruses) that populate the gastrointestinal tract of the animals. Bacteria represent most of the community: recent estimates based on metagenomics indicate that they constitute approximately 98% of the microbiota [1,2] while archaea, eukaryotes, and viruses represent minor components, with percentages between 0.2–1.1% (archaea), 0.4–1.2% (eukaryotes), and 0.1–0.3% (viruses) [2,3]. The intestinal microbiota is implicated in a series of host activities involving intestinal development and functions including epithelial turnover, immune modulation, gastrointestinal motility, and drug metabolism. The bacterial communities also perform important metabolic functions, counteract dietary toxins and carcinogens [4], synthesize micronutrients, ferment indigestible food substances [5,6], assist the absorption of some electrolytes and minerals [7], and influence the growth and differentiation of enterocytes and colonocytes through the production of short chain fatty acids (SCFAs) [8]. Intestinal microbiota is an integral part of the intestinal barrier, which protects the host from invasive pathogens with a mechanism called colonization resistance; proposed mechanisms include competition for oxygen, nutrients, mucosal adhesion sites, and the creation of a physiologically restrictive environment for non-resident bacterial species (e.g., antimicrobial secretion, alterations in gut pH, and production of hydrogen sulfide [9]). However, to have a symbiotic beneficial relationship between microbiota and host, the bacterial species that make up the intestinal microbiota must be in balance with each other to promote a state of homeostasis. The imbalance of intestinal bacterial species has been observed in various canine pathologies, and it is still not clear whether it is the cause or a consequence of them [10,11]. Intense and prolonged sporting activity such as that of sled dogs creates stress capable of compromising the integrity of the intestinal ecosystem, making the animal more susceptible to inflammation and infections.

A state of intestinal dysbiosis is often found in sporting dogs; dysbiosis represents one of the best-known alterations of the intestinal microbiota, able to negatively influence the immune function [11,12]. This condition promotes the destruction of the intestinal barrier with greater access of pathogens into the intestinal lumen, predisposing it to inflammatory reactions [13] with negative consequences not only for the sporting performance of the animal but also for their health. Dysbiosis occurring in the large intestine is typically associated with decreases in major abundant bacterial taxa (e.g., *Blautia*, *Faecalibacterium*, *Ruminococcaceae* and *Turicibacter*), which produce SCFAs, indoles, and other immunomodulatory metabolites with great consequences for host metabolism [14]. To date, various nutritional strategies are used to re-balance the alteration of the intestinal microbiota while the use of antibiotics is increasingly not recommended as they induce a rapid and significant decline in taxonomic richness, diversity, and bacterial uniformity [11]. For this reason, renewed interest has been directed towards pro- and prebiotics and other nutraceutical products. In a recent study [15], it was demonstrated that the use of yeast components as functional ingredients in the canine diet can modulate the immune response, promoting beneficial effects. These components can have protective effects against negative factors that are able to alter intestinal permeability, maintaining the integrity of the intestinal barrier and preventing the passage of potentially pathogenic intestinal bacterial species into the circulation [16]. In the literature, there are several studies that show how the intake of pro- and prebiotics can re-balance an altered microbiota thanks to their synergistic effect [17,18,19]; in fact, it is already known that the potential beneficial effect of prebiotics to improve canine gut health is mediated by changes in the gut microbiota [20]. Garcia-Mazcorro et al. [21] showed how the fructo-oligosaccharides (FOS) consumption increases the amount of specific bacterial orders like *Bacteroidales*, *Bifidobacteriales*, *Lactobacillales*, and *Fusobacteriales*, reducing the number of bacterial species belonging to the *Clostridiales* order. On the other hand, other studies showed how the intake of a probiotic can balance the intestinal ecosystem [22,23]. However, there are no studies on the effects of tyndallized probiotics on the intestinal microbiota of sporting dogs. Tyndallized probiotics are heat-killed bacteria that display the same immunomodulating and antagonizing properties against pathogens of live microorganisms [24,25]. Several studies showed how these dead cells and their metabolites can exert important biological responses promoting the balance of the intestinal microbiota [24,25]. In addition, tyndallized probiotics are safe and do not pose any risk to vulnerable patients [26] where the use of live microorganisms may not be recommended due to the possible translocation of bacteria from the gut to the systemic circulation [24,25,27]. Furthermore, it is very difficult for a live microorganism to colonize an altered intestinal environment such as the one we can find in a dog following intense physical activity. In these cases, the intake of the live bacteria could be favored by an initial introduction of a tyndallized strain or by a nutraceutical product that contains dead bacterial cells which still manage to exert positive effects on the intestinal mucosa by favoring the subsequent adhesion of live bacterial strain. The product analyzed in the present research was already examined in a recently published study; specifically, the study evaluated the ability of Microbiotal cane^®^ to modulate in vitro the intestinal microbiota of healthy dogs, comparing its effects with those exerted by the live probiotic *L. reuteri* DSM 32203 alone [19]. Microbiotal cane^®^ exerts in vitro a short-term effect on the total count of lactobacilli, increasing the concentrations of these beneficial bacterial species during the first hours of fermentation, while the live probiotic alone can reduce the concentrations of bacterial species belonging to the *Bacteroides–Prevotella–Porphyromonas* group. To date, there are still no studies that have evaluated in vivo the effects of a specific postbiotic product on the intestinal microbiota of sporting dogs, which is very interesting as these dogs are already predisposed to an alteration of the microbiota and a product able to re-establish a balanced intestinal condition is certainly necessary.

### Aim of the Study

The aim of the present study is to evaluate the effect of a specific nutraceutical product, Microbiotal cane^®^, on the intestinal microbiota of adult athlete dogs. Specifically, we examined whether this supplement could improve the dysbiotic condition frequently observed following animal training/competitions.

## 2. Material and Methods

### 2.1. Ethic Statement

Ethical review and approval were waived for this study. Because the study protocol requirement for the administration of a nutraceutical is covered by Directive 2010/63/EU of the European Parliament and of the Council of 22 September 2010 on the protection of animals used for scientific purposes, the study is exempt from ethics approval.

### 2.2. Product Characteristics

Microbiotal cane^®^ is the first nutraceutical supplement specifically formulated to counteract intestinal dysbiosis in dogs; thanks to the synbiotic effect of the prebiotic fiber (inulin and fructo-oligosaccharides (FOS)) and the tyndallized probiotic *Lactobacillus reuteri* DSM 32203, it counteracts the proliferation of pathogenic bacteria and improves the intestinal mucosa barrier. In detail, Microbiotal cane^®^ is made up of dried oligofructose (20%), dicalcium phosphate, chicory inulin (14.4%), dried fruit pressing residue (*Citrus sinensis* L. Osbeck, 9.6%), vegetable oils and fats (hydrogenated palm oil and sunflower oil), inactivated bacteria and their parts (*Lactobacillus reuteri* DSM 32203, 8%), sodium pyrophosphate, yeasts (brewer’s yeast), lupine protein flour, salts of organic magnesium acids (stearic acid), sodium chloride, and calcium carbonate. The nutraceutical also contains folic acid (240 mg/kg) and Vitamin B12 (12 mg/kg). One 1.2 g tablet of Microbiotal cane^®^ was administered for every 10 kg of animal weight. Table 1 reports the composition of a Microbiotal cane^®^ tablet.

### 2.3. Animals and Study Design

Twenty healthy adult Alaskan Husky dogs, nine neutered males and eleven spayed females, were enrolled in the study (Table 2). The experiment lasted thirty days: twenty-nine days of training plus the competition day. Animals were randomly divided into two groups: the Microbiotal group, composed of dogs that received daily Microbiotal cane^®^ for the entire length of the study (competition day included), and the control group, composed of dogs that were not administered the nutraceutical product. Each group of animals was made up of ten dogs; specifically, the Microbiotal group was made up of five males and five females (M:F ratio = 1:1) while the control group consisted of four males and six females (M:F ratio = 2:3). The animals enrolled in the study were aged between 2 and 10 years with a body weight between 17 and 28 kg; their body condition was assessed by the veterinarian before the beginning of the study as reported in Table 2. Additionally, fecal score was also evaluated at the beginning and at the end of the 30 study days using a 7-point scoring chart according to the Nestle Purina Fecal Scoring System.

During the experimental period, dogs trained five times a week, alternating resistance and high- (3–6 h/day) and low-intensity (1–3 h/day) exercises. On competition day, the fecal samples were collected from each dog before and immediately after the race to evaluate the effects of the nutraceuticals on the intestinal microbiota. Animal owners were instructed to immediately freeze collected fecal samples and ship them on ice overnight to the laboratory.

### 2.4. Diet

Each dog consumed two main daily meals with the addition of dried tripe-based snacks. The first meal involved the consumption of ‘Barf Artic Sport’, a very appetizing food with a high energy content ready for use by high-level sporting dogs. This complementary food was developed in collaboration with the Department of Animal Nutrition of the Veterinary Faculty of the University of Teramo and the Antartica Italia Racing Team. It is characterized by excellent quality and consists of lamb meat, lamb tripe, trout, pork crackling, fresh eggs, cod liver oil, linseed oil, and yoghurt; Table 3 shows the analytical components of Barf Artic Sport complementary food. The second meal involved the consumption of ‘Monge BWild Low Grain deer’ (Table 4) with the integration of kefir: four spoons in males and three in females, with the addition of half a teaspoon of ‘Golden Paste’ based on turmeric with antioxidant action. In addition, dogs regularly received Condroplus^®^ (Linea Elisir—Pronto Barf, San Cesareo (Roma)), a supplement based on glucosamine and chondroitin sulfate (Table 5). At the end of each training session, a teaspoon of goji berry juice was administered; dogs were not treated with antibiotics in the last three months preceding the study.

The quantity of food administered to each animal was calculated for each dog according to the guidelines of the ‘National Research Council’ (2006) as reported in Table 6; animals had free access to water at any time of the day. Animals did not receive any medications other than Traumasedyl^®^ during the pre-race and post-race trips; this drug was administered during training only to dogs that showed greater fatigue. 

### 2.5. Microbiological Analysis and Dysbiosis Index

DNA was extracted from each canine fecal sample (100 mg) using the MoBio Power Soil DNA Isolation Kit (MoBio Laboratories, Carlsbad, CA, USA) according to the manufacturer’s instructions [28,29,30]. The fecal samples were collected from each group of dogs on the day of the competition at two specific time points: before and after the race. Real-Time PCR tests were initially used to measure the abundances of selected bacterial taxa which, in previous studies, have been shown to be altered in dogs with gastrointestinal diseases [29,31,32,33,34,35,36,37]. Seven bacterial groups were selected to evaluate the dysbiosis extent: *Faecalibacterium* spp., *Turicibacter* spp., *Streptococcus* spp., *E. coli*, *Blautia* spp., *Fusobacterium* spp., and *C. hiranonis* [37]. The degree of dysbiosis was quantified with a single value, called the Dysbiosis Index (DI), which measures the proximity of the sample under examination to the average (prototype) of each class. Therefore, the Dysbiosis Index (DI) is simply defined as the difference between the Euclidean distance between the test sample and the centroid of the ‘healthy’ class and the Euclidean distance between the test sample and the centroid of the ‘diseased’ class. A dysbiosis index less than or equal to zero can be interpreted as normobiosis; the higher the value, the more we are moving away from a condition of normobiosis. Table 7 reports primers and annealing temperature conditions used in the study. All samples were analyzed in duplicate.

### 2.6. Statistical Analysis

Data were analyzed with GraphPad Prism (GraphPad software version 6.01, San Diego, CA, USA). All samples were analyzed in duplicate, and the results are expressed as mean ± standard deviation. Both pre- and post-competition bacterial concentrations were compared within each group of dogs, as well as the values of the dysbiosis index between the two groups of animals (Microbiotal vs. Control). A *p*-value less than 0.05 was considered statistically significant.

## 3. Results

Dogs supplemented with Microbiotal cane^®^ showed different bacterial concentrations compared to the control group; in fact, comparing the results of the quantitative PCR, the seven bacterial groups selected for the calculation of the dysbiosis index show different trends in the two groups of dogs. As regards the control group, *Faecalibacterium* genus shows a significant decrease in the post competition period compared to the pre-competition condition (5.05 ± 0.87 log CFU/mL vs. 4.73 ± 0.92 log CFU/mL (*p* = 0.025); Figure 1); the same trend was observed in other bacterial genera like *Turicibacter*, *Blautia*, *Fusobacterium*, and in *C. hiranonis*, which is the one that had a more evident decline, even if not statistically significant (3.66 ± 1.19 log CFU/mL vs. 2.23 ± 0.82 CFU/mL (Figure 1). The only bacterial genus that showed an increase in the post-competition period was that of *Streptococcus* (6.17 ± 1.15 log CFU/mL vs. 7.05 ± 0.97 log CFU/mL (*p* = 0.0006) (Figure 1), and the same increase was also found in *E. coli* (5.75 ± 1.36 log CFU/mL vs. 6.68 ± 1.15 log CFU/mL (*p* = 0.0002) (Figure 1). 

On the contrary, the group of dogs that took Microbiotal cane^®^ supplement did not show any decrease in bacterial concentrations in the post-competition compared to the pre-competition period (Figure 2). In the post-competition period, a slight increase was recorded only for *Faecalibacterium* and *Streptococcus* spp. (5.47 ± 1.05 log CFU/mL vs. 7.62 ± 1.06 log CFU/mL (*p* = 0.0064); 4.43 ± 0.65 log CFU/mL vs. 4.66 ± 0.99 log CFU/mL (*p* = 0.0088), respectively; Figure 2). *Turicibacter* spp., on the contrary, shows a significant increase between the pre- and post-competition periods (7.09 ± 1.10 log CFU/mL vs. 7.40 ± 1.12 log CFU/mL (*p* = 0.0118); Figure 2). *Blautia* spp., *E. coli*, and *C. hiranonis* show small differences in concentration between the pre- and post-race periods other than a slight increase found in the post-competition period (Figure 2).

The control group shows a higher dysbiosis index in the post-race compared to the pre-race period (3.94 vs. 2.67, *p* < 0.0001; Figure 3); even in the group of dogs that took the postbiotic, the dysbiosis index tends to increase after a competition session compared to the pre-race period (−0.51 vs. −1.31, *p* < 0.01; Figure 3); however, it tends always to remain below zero, the threshold value, maintaining the intestinal ecosystem in a condition of normobiosis. The Microbiotal group shows a significant decrease (*p* = 0.0002) in the dysbiosis index value compared to the control group, as can be seen in Figure 3.

The results obtained from the analysis of the intestinal microbiota are also supported by the fecal score values recorded at the beginning and at the end of the 30 days of study. As can be seen in Table 8, the fecal score tends to increase on the thirtieth day of the study, following the competition, deviating from the value of 2 recorded at the beginning of the study and considered optimal. It was conceivable that following training/competition, effort and stress would cause alterations at the intestinal level associated with the production of softer, moister, and less-formed stools. However, the fecal scores reported in the group of dogs that received Microbiotal cane^®^ appear to be better than those of the control group; although there was a negative change in the consistency of the stool at the end of the race, this change seems to be cushioned in the Microbiotal group by the nutraceutical intake. 

## 4. Discussion

The bacterial taxa analyzed in the present study are fundamental for carrying out the dysbiosis index. Healthy dog fecal microbiota shows the prevalence of *Fusobacterium* spp., *Faecalibacterium* spp., *Turicibacter* spp., *Blautia* spp., *Fusobacterium* spp., and *C. hiranonis*; however, in case of intestinal microbiota alteration, a decrease was observed in the bacterial phyla *Firmicutes* (i.e., *Lachnospiraceae*, *Ruminococcaceae* and *Faecalibacterium* spp.) and *Bacteroidetes*, commonly associated with a concomitant increase in *Proteobacteria* (i.e., *Escherichia coli*) and *Streptococcus* spp. The results obtained from the control group of dogs showed an increase in *Streptococcus* spp. and *E. coli*, considered enteropathogenic to the detriment of beneficial bacterial species such as *Faecalibacterium*, *Turicibacter*, *Blautia*, *Fusobacterium,* and *C. hiranonis*. Also, the group of dogs to which the nutraceutical was administered showed an increase in enteropathogens (*E. coli* and *Streptococcus* spp.), but the increase was smaller than that observed in the control group. Unlike the control group, dogs treated with Microbiotal cane^®^ displayed a greater number of beneficial bacteria belonging to the genera *Faecalibacterium*, *Turicibacter*, *Blautia*, *Fusobacterium*, and *C. hiranonis* in the post-competition period. These bacterial species are considered beneficial as they can help the body in the digestion of food and absorption of nutrients, producing vitamins and substances that bring benefits to the entire organism. Of notable importance is the fact that the alterations found in the control group are the same as those found during acute and chronic intestinal inflammation [35], conditions not observed in the Microbiotal group. In dogs with acute diarrhea, in fact, different studies have shown an increase in the abundance of *E. coli*, *Enterococcus* spp., and *C. perfringens*, associated with a decrease in *Faecalibacterium* spp., *Ruminococcaceae*, and *Blautia* spp. [38,39]; this suggests that dogs suffering from acute intestinal inflammation will present a strong dysbiosis with a probable decrease in short-chain fatty acid (SCFA)-producing bacteria. In chronic enteropathies, a decrease is commonly observed in *Fusobacteria*, *Bacterioidetes*, and *Firmicutes*; furthermore, *Enterobacteria* (indicator of dysbiosis) are over-represented [35]. In association with these alterations, there was also a decrease in *Blautia* spp., *Faecalibacterium* spp., *Turicibacter* spp., and *C. Hiranonis*, while *E. coli* and *Streptococcus* spp. were increased. This correspondence indicates that stress and physical activity play key roles in microbiota alteration.

Physical activity itself induces a stressful situation at the intestinal level with alteration of the microbiota and subsequent inflammatory process, which is highlighted by increased intestinal permeability and diarrheal episodes [40]. Some of these declining bacterial groups are believed to be important producers of various metabolites including SCFAs that are fundamental for intestinal integrity and well-being. In fact, bacteria such as *Faecalibacterium* spp., *Turicibacter* spp., and *Ruminoccocus* spp. ferment dietary carbohydrates to butyrate, acetate, and propionate [41]. These SCFAs also represent a significant source of energy and growth factors for intestinal epithelial cells, act as nutrients that regulate intestinal motility, and create an unsuitable environment for pH-sensitive enteropathogens [42]; additionally, SCFAs display immunomodulatory effects; for example, butyrate induces immunoregulatory T-cells, and acetate effectively modulates intestinal permeability [43]. Therefore, it appears that dysbiosis had a direct impact on the concentration of SCFAs, and this warrants further research into potential therapeutic applications. 

Furthermore, the veterinarian who visited the dogs enrolled in this study found a better fecal consistency in the group of dogs that took Microbiotal Cane^®^ compared to the control group. In fact, after the competition, the fecal samples of the Microbiotal group left residues on the ground once collected and were very moist but still maintained a log shape while the control group showed more soggy fecal samples present in piles rather than logs; these observations were further confirmed by the fecal score values recorded at the end of the study. The differences between the control and Microbiotal group could be related to the higher concentration of *E. coli* and *Streptococcus* spp. found in the fecal samples of dogs belonging to the control group; these bacterial species are often associated with impaired intestinal function.

Additionally, interesting studies showed that Enterotoxigenic *Escherichia coli* (ETEC) is the most common cause of *E. coli* diarrhea in animals and most ETEC isolated from dogs with diarrhea can produce heat-stable enterotoxin A (Sta) and a small proportion of these are also able to release the heat-stable enterotoxin B (STb) [44,45]. Secretion of water and electrolytes in the intestinal lumen results from toxin activity; Enterotoxigenic *Escherichia coli* are known to cause rapid onset of secretory diarrhea leading to dehydration [46]. These bacterial species can colonize and replicate rapidly within the animal intestine, we could hypothesize that an altered intestinal environment such as that which can be found following intense physical activity could promote the proliferation of these pathogenic bacteria, favoring the onset of diarrheal episodes. Enterotoxinemias are caused primarily by *E. coli*, *C. perfringens*, and *C. difficile* and their respective toxins; it might be supposed that these bacterial species are substantially higher in sled dogs with diarrhea than in non-diarrheic dogs during racing. Furthermore, diarrhea associated with *C. perfringens* or *C. difficile* was thought to be exacerbated by coinfection with other putative enteropathogens, including *Campylobacter*, *Salmonella*, *Giardia*, and *Cryptosporidium* spp. [47].

So, the present study found bacterial alterations in the group of dogs that were not administered the nutraceutical product, indicating a possible inflammatory process, as in the case of acute and chronic intestinal pathologies; this condition is only partially found in the group of dogs subjected to nutraceuticals, indicating that Microbiotal cane^®^, even if taken for a relatively short period (30 days), improved the composition of the microbiota during intense physical exercise. Even if there are small increases in *E. coli* and *Streptococcus* spp., the increases observed are less than those in the control group, and at the same time, there is also an increase in beneficial bacteria (*Faecalibacterium* spp., *Turicibacter* spp., *Blautia* spp., *Fusobacterium* spp., and *C. hiranonis*) that favor the production of anti-inflammatory and regulatory metabolites essential for intestinal integrity [48]. This indicates that with a more prolonged treatment, the canine intestinal microbiota conditions would probably progress towards an improvement, even during a competition period, reducing the probability of diarrheal episodes and inflammatory processes.

## 5. Conclusions

From the results obtained in the present study, it can be concluded that Microbiotal cane^®^ could represent an excellent and valid support for sporting dogs as it is able to reduce the negative effects of stress on the intestinal microbiota. Specifically, the postbiotic can maintain a condition of intestinal normobiosis by reducing the growth of potentially harmful bacterial species (*E. coli* and *Streptococcus* spp.), avoiding the decrease in beneficial ones (*Faecalibacterium* spp., *Fusobacterium* spp., *C. hiranonis*). This could be translated into a reduced inflammatory process often associated with physical activity and reduced problems affecting the intestinal compartment such as altered permeability and production of unformed and soft stools; obviously, to confirm this hypothesis, further research is necessary as in the present study, the intestinal microbiota composition and the dysbiosis index were mainly evaluated. Nevertheless, the maintenance of a balanced intestinal ecosystem is fundamental for the general well-being of the host organism as bacterial species that inhabit the intestinal compartment can release metabolites capable of interacting not only locally but also with distant organs, so the conservation of a healthy intestinal microbiota is very important and Microbiotal cane^®^ seems to promote this process. However, the study evaluated a short period of postbiotic intake (30 days), and it would be interesting to see whether a more prolonged intake over time could be able to enhance the positive effects already recorded.

## Figures and Tables

**Figure 1 animals-14-02226-f001:**
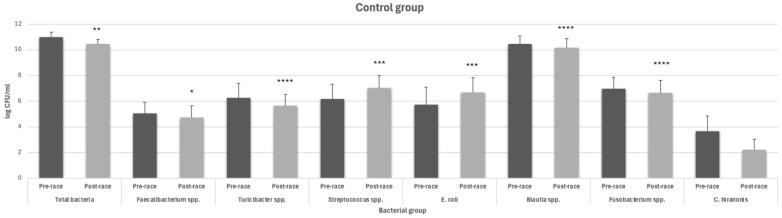
Bacterial concentration expressed as log UFC/mL in the pre- and post-training period in the control group (* significantly different *p* < 0.05, ** *p* < 0.01, *** *p* < 0.001, **** *p* < 0.0001 with respect to the pre-competition period).

**Figure 2 animals-14-02226-f002:**
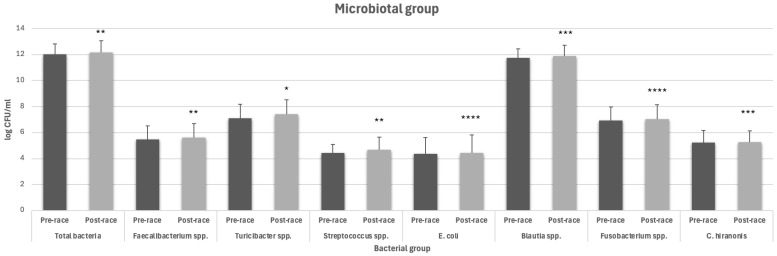
Bacterial concentration expressed as log UFC/mL in the pre- and post-training period in the Microbiotal group (* significantly different *p* < 0.05, ** *p* < 0.01, *** *p* < 0.001, **** *p* < 0.0001 with respect to the pre-competition period).

**Figure 3 animals-14-02226-f003:**
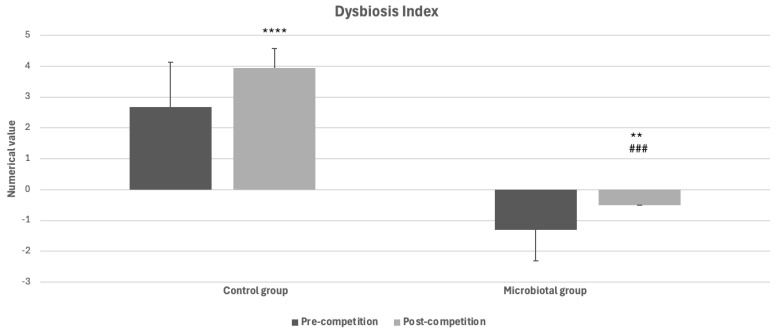
Dysbiosis Index calculated in the Control and Microbiotal groups of dogs before and after a training session (** *p* < 0.01, **** *p* < 0.0001 with respect to the pre-competition period; ^###^ significantly different (*p* < 0.001) with respect to the post-competition period of the control group).

**Table 1 animals-14-02226-t001:** Composition of Microbiotal cane^®^ tablet.

Components	Quantity (mg)
FOS	250.00
Inulin	180.00
Tyndallized *Lactobacillus reuteri* DSM 32203	100.00
Polyphenols	120.00
Microencapsulated butyric acid	200.00

Composition for one tablet (1.2 g).

**Table 2 animals-14-02226-t002:** Age, sex, and body weight of dogs enrolled in the study. (**A**) Control group, (**B**) Microbiotal group.

**(A) Control Group**				
**Dogs**	**Age (Years)**	**Sex**	**Body Weight (kg)**	**Body Condition Score (BCS)**
Damon	7	M	28.00	4
Deah	5	F	21.40	3
Fiona	2	F	22.50	4
Quest	10	M	22.90	4
Nebula	2	F	22.60	4
Grace	2	F	20.00	4
George	2	M	24.30	4
Double	5	F	20.00	4
Domitilla	5	F	20.00	3
Fangio	2	M	25.30	3
**(B) Microbiotal Group**				
**Dogs**	**Age (Years)**	**Sex**	**Body Weight (kg)**	**Body Condition Score (BCS)**
Gaston	8	M	26.40	3
Jolie	7	F	19.10	5
Desire	5	F	20.50	4
Anastasia	9	F	22.20	4
Gail	2	M	23.60	3
Alex	9	M	26.50	4
Griffith	2	F	20.40	3
Smokie	4	M	20.10	4
Ivon	7	F	17.90	3
Ferguson	2	M	25.30	4

**Table 3 animals-14-02226-t003:** Analytical components of Barf Artic Sport.

Analytical Components	Percentage (%)
Dry matter	62.40
Crude protein	24.90
Crude fats	29.50
Raw ash	1.90
Carbohydrates	6.20
Raw cellulose	0.01
Metabolizable energy (Kcal/Kg)	3800.00

**Table 4 animals-14-02226-t004:** Analytical components of Monge BWild Low Grain deer.

Analytical Components	Percentage (%)
Crude protein	28.00
Raw fiber	2.60
Crude fats	18.00
Raw ash	7.50
Calcium	1.60
Phosphorus	1.00
Omega 3 fatty acids	0.50
Omega 6 fatty acids	1.70
Metabolizable energy (Kcal/Kg)	4170.00

**Table 5 animals-14-02226-t005:** Condroplus^®^ composition.

Analytical Components	Percentage (%)
Crude protein	9.00
Crude fats	1.22
Raw fiber	0.30
Raw ash	6.00

Nutritional additives: Vitamin E (27,800 mg) and Vitamin C (50,000 mg). How to use: 1 tablet every 10 kg of weight.

**Table 6 animals-14-02226-t006:** Energy requirement and quantity of feed (g) daily consumed by dogs in the Control (**A**) and Microbiotal group (**B**) of dogs.

**(A) Control Group**			
**Dogs**	**Energy Requirements**	**BWild (g/d)**	**Barf Artic Sport (g/d)**
Damon	1357.00	340.00	350.00
Deah	1040.00	261.00	250.00
Fiona	1098.00	275.00	250.00
Quest	1193.00	299.00	350.00
Nebula	1136.00	285.00	250.00
Grace	1040.00	261.00	250.00
George	1267.00	317.00	350.00
Double	1079.00	270.00	250.00
Domitilla	981.00	246.00	250.00
Fangio	1357.00	340.00	350.00
**(B) Microbiotal Group**			
**Dogs**	**Energy Requirements**	**BWild (g/d)**	**Barf Artic Sport (g/d)**
Gaston	1303.00	326.00	350.00
Jolie	1001.00	251.00	250.00
Desire	1060.00	266.00	250.00
Anastasia	1136.00	285.00	250.00
Gail	1193.00	299.00	350.00
Alex	1303.00	326.00	350.00
Griffith	1001.00	251.00	250.00
Smokie	1136.00	285.00	350.00
Ivon	981.00	246.00	250.00
Ferguson	1230.00	308.00	350.00

**Table 7 animals-14-02226-t007:** Primers and annealing temperatures.

qPCR Primers	Sequence (5′–3′)	Target	Annealing (°C)
Forward	GAAGGCGGCCTACTGGGCAC	*Faecalibacterium*	60
Reverse	GTGCAGGCGAGTTGCAGCCT		
Forward	GGGCTCAACMCMGTATTGCGT	*Fusobacteria*	51
Reverse	TCGCGTTAGCTTGGGCGCTG		
Forward	TCTGATGTGAAAGGCTGGGGCTTA	*Blautia*	56
Reverse	GGCTTAGCCACCCGACACCTA		
Forward	CCTACGGGAGGCAGCAGT	Total bacteria	59
Reverse	ATTACCGCGGCTGCTGG		
Forward	CAGACGGGGACAACGATTGGA	*Turicibacter*	63
Reverse	TACGCATCGTCGCCTTGGTA		
Forward	GTTAATACCTTTGCTCATTGA	*E. coli*	55
Reverse	ACCAGGGTATCTAATCCTGTT		
Forward	AGTAAGCTCCTGATACTGTCT	*C. hiranonis*	50
Reverse	AGGGAAAGAGGAGATTAGTCC		
Forward	TTATTTGAAAGGGGCAATTGCT	*Streptococcus*	54
Reverse	GTGAACTTTCCACTCTCACAC		

**Table 8 animals-14-02226-t008:** Fecal Score values recorded before and at the end of the 30 days of study in the Control (**A**) and Microbiotal group (**B**) of dogs.

**(A) Control Group**		
**Dogs**	**Fecal Score—Beginning of the Study**	**Fecal Score—End of the Study**
Damon	2	6
Deah	2	6
Fiona	2	6
Quest	2	5
Nebula	2	6
Grace	2	6
George	2	5
Double	2	6
Domitilla	2	5
Fangio	2	5
**(B) Microbiotal Group**		
**Dogs**	**Fecal Score—Beginning of the Study**	**Fecal Score—End of the Study**
Gaston	2	4
Jolie	2	4
Desire	2	4
Anastasia	2	4
Gail	2	4
Alex	4	6
Griffith	2	4
Smokie	2	4
Ivon	2	4
Ferguson	2	4

## Data Availability

The original contributions presented in the study are included in the article, further inquiries can be directed to the corresponding authors.

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
