# Peer review of "Effects of a Nutraceutical Treatment on the Intestinal Microbiota of Sled Dogs"

_animals, 2024, doi:10.3390/ani14152226_

Round 1

Reviewer 1 Report

Comments and Suggestions for Authors

The study is well designed, reported and conclusions are good and balanced. Information that I miss is the dog breed, I guess Alaska huskies, and the training performance and intensity in the 30 days period. To compare with other studies this is important to know. Also energy intake and body weight development in the experimental period would have been interesting to add if that was registered.    

Author Response

First of all, thank you for your comments on the study.

I have included more information regarding the bodyweight, body condition and energy intake of each dog enrolled in the study.

Reviewer 2 Report

Comments and Suggestions for Authors

Effects of a Nutraceutical Treatment on the Intestinal Microbiota of Sled Dogs

**please see PDF of paper with additional comments and suggested amended/expansions/clarifications**

Overall comments

Thank for you the opportunity to review this interesting and relevant paper exploring the impact of a commercially available supplement for supporting digestive health, especially in sporting dogs.

This is a current area of interest and relevant in terms of performance, health and welfare.

Additional specific, valid and verified evidence is needed to support the use of nutraceuticals and similar supplements, especially for digestive health which is a key concern for many canine caregivers.  For this reason, I consider this work of value to the wider field, but I temper my enthusiasm for this work by cautioning against extrapolation of results to significance and linking to inflammatory modulation without sufficient supporting evidence to back this up.

This is a research article and considers the background to study well, including setting the scene as to challenged faced by sporting and performance dogs. I would like more supporting evidence in many areas – highlighted on the script.

Study design and protocol appear robust and appropriate, but I feel there is information missing that is warranted in terms of materials and methods;

·       More study detail on dog recruitment – was this a single kennel? Details on exercise levels and standard activity regimens?

·       Nutritional detail is a little lacking

·       Only microbiota examined – what about canine performance and other criteria? Bodyweight? Any other visible criteria that can be scored?

·       More detail on molecular techniques used including protocols for replication.

·       Ethical review??

The premise of the work is clear, and the undertaking and analysis appears mostly appropriate but there are areas I feel warrant review and additional development.

Overall structure, flow and language use is clear and supports readability but there could be enhanced paragraph use. Comments are detailed within the manuscript for specific consideration.

Overall, I feel the work is robust and solid but does have scope to further develop some areas of description and discussion, including avoidance of extreme extrapolation based on a small sample and limited experimental assessment.

Keywords – relevant and appropriate

Figures and tables are fine and suitable for the work.

References

I have not exhaustively gone through these, but all appear fine – present and correct although I have not proofread or cross referenced to check validity of use.

Comments on the Quality of English Language

Author Response

I revised the manuscript and thanks to the careful advice given I proceeded to add additional details regarding bodyweight, body condition score, nutritional and ethical information regarding the dogs enrolled in the study.

Reviewer 3 Report

Comments and Suggestions for Authors

Summary of work in manuscript

            This study examined effects of a commercial pro/prebiotic supplement (Microbiotal cane) before and after a sled race in sled dogs supplemented for 30 days on gut microbiome and gut dysbiosis index. The study is adequately designed, but the statistics do not seem appropriate. Moreover, the writing needs some English editing and much of the discussion is speculative, unsupported and/or lacks appropriate referencing. Specific issues for the authors to remedy are:

1.    Line 20: Delete ‘and’

2.    Line 21: What does ‘and integration’ mean? Clarify what is the authors intend here.

3.    Line 21: Replace ‘carried out at its best’ with ‘optimal’.

4.    Line 22: Replace ‘implies’ with ‘induces’

5.    Lines 29-31: The last few sentences of this abstract are all results. There is no concluding sentence. What do the authors recommend for use of this nutraceutical in preventing exercise-induced intestinal dysbiosis in sled dogs?

6.    Line 55: change to ‘…balance with each other…’

7.    Line 86: NBF1 is an undefined acronym

8.    Line 102: Define this acronym FOS. Moreover, FOS and inulin have not been introduced yet. Might be worth a mention in the Intro by adding a sentence or two with suitable references.

9.    Section 2.2: Were the dogs intact, spayed or neutered?

10. Lines 128-129: What was the last day of the experiment? Was it day 30 or was it after day 30? Please specify here in the methods text.

11. Figures 1 and 2: Why did the authors choose to analyze the data shown in Figures 1 and 2 separately? In doing so, the most critical comparison to this study, comparing between control and treatment groups, is not possible. Thus, as is, the authors are unable to justify saying that there are differences between control and treatment groups in this study.

12. Figure 3: Where is the statistics for this figure? A 2-way ANOVA seems the most appropriate, to enable statistical testing of whether treatment or racing have significant effects on this dysbiosis index. If there are not any significant overall effects for these factors, then the authors cannot claim there are differences in this index.

13. Line 233: The authors should temper this statement by acknowledging that they did not directly measure SCFA in this study. They can only infer that because bacteria known to produce SCFA were decreased, they would predict a decreased intestinal fermentation and lower SCFAs.

14. Lines 253-254: There is an overly large discussion of intestinal fermentation and effects of SCFAs. Considering that this study did not measure SCFAs, I would suggest limiting most of this discussion. Was diarrhea observed in this study during the race? The authors should attempt to link the pathology-induced changes in fecal consistency to what was observed in this study.

15. Lines 261-266: This seems highly speculative and no reference is provided. Since the authors did not measure any of the hormones or splanchnic blood flow in this study, I suggest that this be deleted.

16. Lines 266-268: Please provide a reference for the statement that these bacteria cause diarrhea through production of enterotoxins. Moreover, the authors need to better explain why they think that the racing dogs would have higher enterotoxin levels. Again, they did not directly measure toxin levels in the feces. Did they do any metagenomic analyses? This seems highly speculative otherwise.

17. Lines 291-293: Either delete this sentence or acknowledge that these are speculative benefits that were not directly measured in this study.

Comments on the Quality of English Language

See above for specific corrections to grammar that are needed. The whole manuscript should be checked again.

Author Response

I proceeded to attach the file with all the answers to the questions.

Round 2

Reviewer 2 Report

Comments and Suggestions for Authors

Thank you for reviewing and editing this paper - Overall, i believe it to be more robust and detailed overall with enhanced clarity and detail.

Two comments - I would suggest avoid using the term 'prevent' in relation to the use of this nutraceutical - prevent is viewed as a drug/pharmaceutical claim in my country so could be problematic in commercial use.

Second comment - are the components of the nutraceutical regulated feed additives? Are they on the FAR/FMR - might be good to note (or not)

Reviewer 3 Report

Comments and Suggestions for Authors

The authors have inserted a large amount of new data and tables with corresponding text which greatly improves this manuscript. The authors have addressed all concerns this reviewer had and I have nothing further to suggest.